Deep ensemble learning for gastrointestinal diagnosis using endoscopic image classification

Siddiqui Samra 1
Khan Junaid Ali 1
http://orcid.org/0000-0003-3435-6681 Algamdi Shabbab 2 s.algamdi@psau.edu.sa
1 Department of Computer Science, HITEC University , Taxila , Pakistan
2 Department of Software Engineering, College of Computer Science and Engineering, Prince Sattam bin AbdulAziz University , AlKharj , Saudi Arabia
Chaki Jyotismita
Electronic publication date: 2025 Apr 22
Publication date: 2025
Volume: 11
Electronic Location ID: e2809
Received 2024 Dec 17; Accepted 2025 Mar 16
Copyright: © 2025 Siddiqui et al.
Copyright year: 2025
Copyright holder: Siddiqui et al.
License: This is an open access article distributed under the terms of the Creative Commons Attribution License, which permits unrestricted use, distribution, reproduction and adaptation in any medium and for any purpose provided that it is properly attributed. For attribution, the original author(s), title, publication source (PeerJ Computer Science) and either DOI or URL of the article must be cited.
License URL: https://creativecommons.org/licenses/by/4.0/

Keywords: Transfer learning, Deep convolutional neural networks, Ensemble learning, Gastrointestinal disease, NasNet mobile

Funding: Prince Sattam bin Abdulaziz University PSAU/2025/R/1446 The authors received no funding for this work. The APC was supported via funding from Prince Sattam bin Abdulaziz University, project number (PSAU/2025/R/1446). The funders had no role in study design, data collection and analysis, decision to publish, or preparation of the manuscript.

==============================
Transfer learning is a valuable tool for the effective assistance of gastroenterologists in the powerful diagnosis of medical images with fast convergence. It also intends to minimize the time and estimated effort required for improved gastrointestinal tract (GIT) diagnosis. GIT abnormalities are widely known to be fatal disorders leading to significant mortalities. It includes both upper and lower GIT disorders. The challenges of addressing GIT issues are complex and need significant study. Multiple challenges exist regarding computer-aided diagnosis (CAD) and endoscopy including a lack of annotated images, dark backgrounds, less contrast, noisy backgrounds, and irregular patterns. Deep learning and transfer learning have assisted gastroenterologists in effective diagnosis in various ways. The goal of proposed framework is the effective classification of endoscopic GIT images with enhanced accuracy. The proposed research aims to formulate a transfer learning-based deep ensemble model, accurately classifying GIT disorders for therapeutic purposes. The proposed model is based on weighted voting ensemble of the two state-of-the-art (STA) base models, NasNet-Mobile and EfficientNet. The extraction of regions of interest, specifically the sick portions, have been performed using images captured from endoscopic procedure. Performance evaluation of the proposed model is performed with cross-dataset evaluation. The datasets utilized include the training dataset HyperKvasir and two test datasets, Kvasir v1 and Kvasir v2. However, the dataset alone cannot create a robust model due to the unequal distribution of images across categories, making transfer learning a promising approach for model development. The evaluation of the proposed framework has been conducted by cross-dataset evaluation utilizing accuracy, precision, recall, Area under curve (AUC) score and F1 score performance metrics. The proposed work outperforms much of the existing transfer learning-based models giving 97.83% on Kvasir v1 and 98.45% accuracy on Kvasir v2.

Introduction

Gastrointestinal disorders (GIDs) have a good chance of significantly affecting a wide range of individuals at a substantial rate. Research by the World Health Organization indicates that more than 1.8 million individuals face severe challenges and ultimately succumb to gastrointestinal disorders annually (Hmoud Al-Adhaileh et al., 2021). The diagnosis and elimination of GIT illnesses is considered an evolving field of research. Some common stomach disorders include esophagitis, bleeding, polyps, and ulcers (Sharif et al., 2021). Globally, stomach cancer ranks third in terms of its significant contribution to mortality (Lee et al., 2019). Similarly, esophageal cancer is ranked sixth among all types of cancer (Ghatwary, Ye & Zolgharni, 2019). In the year 2021, a total of 26,560 instances of stomach cancer were reported, comprising 1,600 cases in males and 1,040 cases in females (Strøbech, Giuriatti & Erler, 2022). The estimated number of patients diagnosed with gastric cancer in the United States in 2022 was 2,600. There were 1,500 reported cases in men and 1,040 reported in women. Of these cases, the death rate reached 11,090 with 6,690 male and 4,400 female cases (Adadi, Adadi & Berrada, 2019).

In recent years, advances in Artificial Intelligence (AI) and computer vision (CV) have significantly improved the accuracy of classification of GIT disorders. Images must first be pre-processed, but it is an inevitable and challenging task due to the presence of annotations, variations in background and light intensity, and other artifacts (Khan et al., 2019). The level of detail of endoscopic images received during a patient’s examination significantly affects the accuracy of disease detection (Ramamurthy et al., 2022).

Computer-aided diagnosis (CAD) techniques can benefit radiologists in interpreting images and diagnosing diseases. Previous studies have employed machine learning techniques, including naive Bayes and random forest, for endoscopic image classification (Naz et al., 2021; Adadi, Adadi & Berrada, 2019). The percentage accuracy of the utilized technique is highly dependent upon the features that are extracted for model development. Moreover, deep neural networks (DC-NNs) such as ResNet (Sandler et al., 2018), VGG-16 (Muruganantham & Balakrishnan, 2021), Google-Net (Strøbech, Giuriatti & Erler, 2022) and other pre-trained models have enabled automated classification with high accuracy–thus, reducing the burden on healthcare professionals and gastroenterologists. Transfer learning (TL) involves utilizing a pre-trained classification model that includes frozen layers to extract features. The act of freezing some layers is performed as an aspect of fine-tuning, to reduce both training and computing costs. The previous methods used multiple pre-trained models to accurately diagnose gastrointestinal tract problems. The accuracy of the acquired findings is contingent upon the type of pre-trained models employed and the quantity of the dataset utilized for both training and testing.

Despite recent advances in the medical field, the accurate diagnosis of GIT remains a challenging task. Although highly effective, traditional diagnostic methods, such as endoscopy, are invasive and more expensive and may not always be practically applicable for screening procedures. In addition, these techniques require trained professionals. Thus, it creates a compelling need for the development of efficient and cost-effective methods for early detection.

Recent advances in modern imaging techniques present promising solutions that could revolutionize the way GI diseases are detected and diagnosed. Here, in the proposed research work, we formulate a novel deep ensemble model of the two base models of the state-of-the-art (STA), NasNet-Mobile and EfficientNet, for a more accurate and early diagnosis. NasNet-Mobile is an architecture optimized using neural architecture search (NAS) to find the best-performing lightweight CNN for mobile devices. The NAS optimization process focuses on building efficient models that achieve high accuracy while maintaining a low computational footprint. EfficientNet, on the other hand, is designed using a compound scaling method that uniformly scales the depth, width, and resolution of the network to achieve optimal accuracy with minimal computation. This approach allows EfficientNet to outperform many traditional architectures, such as ResNet and Inception, while being more computationally efficient. Its ability to achieve superior accuracy while using fewer parameters makes it an ideal candidate for ensemble learning, especially in resource-constrained environments. Both NasNet-Mobile and EfficientNet are state-of-the-art deep learning models designed to maximize the balance between accuracy and computational efficiency, making them well-suited for inclusion in an ensemble model.

To enhance the performance of GIT classification, we propose an ensemble model that combines the strengths of NASNet and EfficientNet. NASNet, known for its efficient architecture and powerful feature extraction capabilities, is paired with EfficientNet, which utilizes a compound scaling method to achieve state-of-the-art results with fewer parameters. By integrating these two models, we aim to leverage their complementary strengths, improving both classification accuracy and computational efficiency. The ensemble approach allows for more robust decision-making by combining the diverse feature representations learned by both networks. This hybrid model addresses the limitations of individual architectures, offering a more comprehensive solution to the complexities of GIT classification. The motivation behind this method is to exploit the advantages of both models while mitigating their individual weaknesses, ultimately advancing the performance of deep learning systems in medical image analysis. The ensemble model combines the most discriminant features of the two base models. This could lead to a significant overfitting problem, but is resolved by adjusting the values of the hyperparameters, resulting in improved percentage accuracy. The primary contributions of the proposed work are enumerated below: 1. We propose a novel deep ensemble network model to improve the classification accuracy of GIT images. The goal of proposing a customized model is to smooth out the overfitting problem and to better generalize on unseen data compared to other pre-trained models.

2. We formulate a new strategy for introducing heterogeneity in data along with handling imbalanced datasets commonly seen in medical diagnosis, where certain disorders are underrepresented. Addressing data imbalance with our proposed model leads to more reliable model predictions.

3. To make the model more resilient to noise, the HyperKvasir dataset is used for training. With the proposed framework, the accuracy is 98.45%, which considerably eliminates constraints including dataset sizes and classification accuracy.

The article begins with an introduction section, followed by “Literature Review”, which provides a review of diagnostic procedures for gastrointestinal diseases and elucidates the limitations associated with current methodologies. “Materials and Methods” provides a comprehensive analysis of the datasets utilized, describes the architecture model in detail and the pre-processing and fine-tuning procedures used for accuracy and performance improvement. “Results and Discussion” presents the findings and their further analysis. The research effort is concluded in “Conclusion and Future Work”, which identifies future prospective research areas in the automated diagnosis of gastrointestinal disorders.

Literature review

Gastrointestinal diseases refer to illnesses that predominantly affect the digestive system, including the upper and lower GIT. The upper tract includes the esophagus, stomach, duodenum, and some part of the small intestine, while the lower tract includes the remaining part of the small intestine, ileum, large intestine, rectum, and colon. The severity of the illness and the intensity of the symptoms may range from severe to mild and can impact people of all age groups. Previous studies have shown many deep models to achieve the precise diagnosis of GIT diseases. One of the most frequently referenced academic articles in this field relates to the study of gastrointestinal cancers and their correlation with human health (Jardim, de Souza & de Souza, 2023). Provided an in-depth review of malignancies that affect the digestive system, it covers their identification, classification, clinical manifestations, and genetic modifications. The research offered an in-depth analysis of the anatomical structure of the GIT, including its tumor type and the associated subtypes, based on molecular traits and characteristics. The authors in Peyrin-Biroulet et al. (2015) have provided a comprehensive explanation of the revised classification paradigm for inflammatory bowel disorders (IBD).

Similarly, Thoeni (2015) offered information on the most recent imaging modalities used to classify various disorders of the human digestive system. Devi et al. (2025) proposed a novel model based on attentional deep learning for GIT segmentation. Moreover, Özbay (2024) used Residual Inception Transformer for GIT segmentation. These disorders need to be treated well in time to save life of millions. The study of technological frameworks has witnessed a surge in research due to recent breakthroughs. Table 1 provides a comprehensive summary of the research conducted on the application of ensemble learning for the accurate classification of GIT.

Table 1 Transfer learning frameworks for classification.

Year	Ref.	Network	Method	Performance	Dataset	
2024	Wei, Mchugh & Mooney (2024)	Pretrained TL	Early epilepsy diagnosis by using pretrained models based on TL	Acc: 0.77, F1: 0.85, Bal. Acc: 0.77	Private: 350 normal, 597 abnormal EEG	
2024	Almakky et al. (2024)	Deep learning	Kernel-based feature fusion on Skin diseases	Acc: 0.814, F1: 0.714	Skin, CheXpert, RSNA Pneumonia	
2022	Oukdach et al. (2022)	Deep learning	Fine-tuning pre-trained network model	Acc: 0.92, F1: 0.87	Kvasir (5,000 images)	
2022	Raut & Gunjan (2022)	Deep learning	TL-based video summation procedures	Acc: 0.92, F1: 0.87	Kvasir V2 (8,000 images)	
2020	Bamne et al. (2020)	TL	Feature-based diagnosis	Prec: 0.88, Acc: 0.90	CUB 200-2011	
2020	Escobar et al. (2020)	TL	Features extracted with transfer learning and CNN	Accuracy: 0.94, Precision: 0.9	8,000 images of dataset Kvasir V2	
2020	Amina, Nadjia & Azeddine (2021)	CNN	Pretrained VGG-16 based proposed framework for image categorization with transfer learning	Accuracy: 0.96	i. CVC-Clinic DB ii. Kvasir V1 with 5,000 images.	

Studies suggest that with increasing number of training images and dataset size, the results become more accurate. Gabor capsule network was proposed for the classification of complex image sets such as the Kvasir dataset with 91.50% accuracy (Abra Ayidzoe et al., 2021). GIT diseases were classified into eight classes with an accuracy of 93.65% using wavelet transform (Mohapatra et al., 2021). However, certain factors can degrade the performance of CNN architecture, including low-contrast video frames, the selection of multiple parameters, and the arrangement of multiple layers in the architecture. By integrating multiple base models, the proposed model addresses these challenges faced by providing a lightweight, reliable model with accuracy gains.

In conclusion, GIT disorders detection and classification is a vital medical research domain and various research work is being carried out in this field. These research articles mainly focus on multiple individual models used for classification along with their ensemble models for improved accuracy and performance. These articles describe how the ensemble models outperform various individual models used for feature identification and classification based on their clinical characteristics. These detection and classification systems aid in the clinical treatment of patients with GIT disorders.

Materials and Methods

The proposed model consists of four major steps. Figure 1 depicts the complete workflow of the proposed model. The process starts with image preprocessing operations as HyperKvasir is a largely unbalanced dataset. Then, input images are subjected to the base models with different weights. Then an ensemble model is created on the basis of weighted averaging ensemble technique. The detailed stages of the whole methodology for GIT classification are presented here.

Figure 1 Workflow of the proposed architecture, utilizing EfficientNet (Source Ahmed & Sabab, 2022) and NasNet-Mobile (Source Karagiannakos, 2021) trained on the HyperKvasir dataset for GIT image classification.

The predictions represent eight classes of Kvasir dataset (Source Pogorelov et al., 2017).

Image preprocessing

Training images are extracted from the HyperKvasir dataset. HyperKvasir has an unequal distribution of images in various categories. These images vary in terms of size and illumination. These images need to be resized for efficient feature extraction. Efficient feature extraction leads to better results with greater classification accuracy. The images of the dataset range from 720×576 pixels to 1,920×1,072. Raw data have been resized to the desired scale for dataset preprocessing and augmentation operations. Execution time is greatly impacted by the Image sizes. Thus, to reduce the execution time, we resized the images of HyperKvasir to 224×224.

As HyperKvasir is an unequal dataset, So, The problem of small training images in different categories is resolved by carrying out augmentation operations (Nouman Noor et al., 2023). Data augmentation is performed on all training categories of HyperKvasir, to get more sample images for training. Rotation, x-shearing, x and y-translation, and y-reflection operations are performed to greatly increase the number of training samples. Flip operations are mathematically represented by the equation mentioned in the following.

Suppose that the image matrix of size 256×256 is expressed as Is,t with s rows and t columns and Is,t∈Ks×t. The image row s=1,2,3,…χ and column t=1,2,3,…ρ provided images are RGB three channeled. The augmentation procedures mentioned above are applied to the sample images and the transpose is given by Eq. (1) and is represented by IT. The transpose operation alters the image indices.

(1) IT=It,s.

Horizontal flip IHF is given by Eq. (2).

(2) IHF=Iρ+1−s.

IVF illustrates the vertical image flip which is given by Eq. (3).

(3) IVF=Iχ+1−t.

Transfer learning

In medical research, a robust classification model can be built using deep learning (Agrawal et al., 2017), but this task can be more challenging due to the smaller number of available sample images. In such scenarios, applying transfer learning can yield good results. Transfer learning is the use of knowledge from one task to help perform a similar task (Zhou et al., 2021; Weiss, Khoshgoftaar & Wang, 2016). For small datasets, creating completely new models from scratch might result in overfitting problems or a model with generalization errors (Agrawal et al., 2017). In some cases, the sample images are unequally distributed in different classes. Transfer learning helps in such scenarios to get an efficient model with greater accuracy and minimal computation time (Berzin et al., 2020).

In the proposed work, our deep ensemble model was trained with HyperKvasir dataset. The proposed architecture network model characterizes these steps. Data augmentation (mirror flipping, horizontal and vertical) is applied to both Kvasir v1 and v2 datasets to increase dataset size. The input data are then fed into the proposed model based on transfer learning and the two best-performing networks are selected. At last, features are extracted from the two best-performing networks, and an ensemble model is created.

Proposed model architecture

Ensemble learning has played a vital role in machine learning and computer vision. Various classifiers are constructed on the basis of multiple ensemble strategies for the resolution of computationally intensive real-world situations and complex problems. The primary goal is an improved ensemble model with considerably optimum results on the basis of performance metrics for effective classification.The workflow steps of the ensemble model can be depicted in the Fig. 2.

Figure 2 Proposed ensemble model architecture for classifying gastrointestinal tract disorders using pre-trained classifiers.

The process starts with the selection of the individual classifiers that give better classification results. The selected learners are effectively combined through a specific strategy; after the learners’ training, the results are analyzed to reach conclusions. The individual classifiers are trained on some pre-trained models along with some frozen layers and defined parameters. Training models with multiple instances of the same individual classifiers come under homogeneous ensemble, while heterogeneous ensemble learning involves training with different individual classifiers. We have used homogeneous ensemble learning in the proposed research, as it involves the combination of two classifiers, which falls under neural networks.

For reducing time and space redundancy, advanced ensemble deep learning frameworks are utilized for advanced medical diagnosis, but this might create a problem of accuracy in real-world scenarios (Yang et al., 2020, 2021; Li et al., 2021). To address these accuracy constraints for improved classification accuracy and better results, the adoption of more efficient ensemble learning techniques is inevitable. The most common ensemble-based methodologies include voting-based selection, averaging-based selection, and stacking-based framework. The voting-based framework is generally used for the classification of binary medical images. The voting-based framework assumes m different classifiers l1,l2,l3,…lm that predict the category from the c category markers t1,t2,t3,…tc based on the classifier outcomes. The predicted result of the classifier denoted by li, for every s sample, is a 1D (li1(s),li2(s),li3(s),......lic(s))m, labeled vector, where lij(s) represents the predicted outcome provided by the li classifier for each tj class.

We have preferred a weighted voting methodology over many existing frameworks considering the importance of usability and the existing complexity of the system. The weighted voting strategy is better compared to the individual base learners as multiple classifiers are compared, and based on the best results, one of the classifiers is selected. Equation (4) gives the formula for these multiple classifier selections.

(4) H(x)=targjmax∑i=1mwilij(s)

where wi denotes the weight. In real world applications, weight coefficients w needs to be normalized and ∑i=1mwi=1. The vital part lies in selecting the right weight value. Let us suppose the sum of the result of the classifier is represented as c=(c1,c2,c3,c4,…cm)m as the individual output, where ci denotes class prediction with label li on a dataset of sample size s. Suppose ρi denotes the precision value of li. Then, the consolidated output of label tj is represented by Eq. (5) by using a Bayesian optimal discriminant function:

(5) Lj(s)=log⁡(B(tj)B(c|tj)).

In the proposed research, an ensemble framework, consisting of two individual pre-trained base learners is used: NasNet-Mobile and EfficientNet model. Assuming the fact that the individual base learner outputs are independent, Eq. (6) can be derived from Eq. (5):

(6) Lj(s)=log⁡(B(tj)+∑i=1mlij(s)log⁡(ρi1−ρi).

In Eq. (6), when log⁡(B(tj)) does not depend upon the results from individual base learners, the optimized selected weight value meets the criteria provided by the below-mentioned Eq. (7):

(7) wi∝log⁡(ρi1−ρi).

As the outputs of all individual classifiers do not depend on other classifiers, the selected optimum weight value should be well aligned with each classifier. In this proposed research, individual base learners are learned for the same set of problems. The outcomes obtained were strongly related making the independence assumption invalid. Therefore, the weight values are adjusted according to the evaluation criteria of all classifiers. We have set weight value wt1 of NasNet-Mobile at 0.6 and wt2 of EfficientNet model at 0.4. We have performed a hyperparameter optimization cross-validation process that revealed that NASNet-Mobile leads to a more accurate combined prediction, thus being assigned a higher weight compared to EfficientNet. The goal of cross validation process is to maximize the performance of each individual model so that when they are combined into an ensemble, they perform optimally.

The initial basic layers carry information about the very basic features, so they are not trained to reduce training time. After the basic layers, the top layers that carry specialized features are subjected to training, so they will learn the set of specialized features. The top layers are then followed with several max-pooling layers. Then comes the dense layers of models that are made up of 1,024, 512, 256, and 128 neurons, respectively. An activation function, ReLU activation, is used and then batch Normalization layer comes with a dropout factor of 0.5. This is concluded with a Fully Connected (FC) layer appended at the end and an eight-neuron output layer because of a total of eight classes in Kvasir v1 and v2 datasets.

Some individual base learners may perform well in some cases but not in all scenarios. With the merger of various learner models, the knowledge gained from all models is combined to get enhanced output. The proposed deep learning-based ensemble model combines the features extracted from various models and thus performs well in all situations. Algorithm 1 gives a detailed algorithm of the proposed approach.

Algorithm 1 Transfer learning algorithm of proposed approach.

Input: Two individual base learners pre-trained on ImageNet, dataset D, and the upper bound of iterations M.	
1. Initialize weight wstr as the weight vector from the t-th neuron in the (r−1)-th layer to the s-th neuron in the r-th layer. Set activation Bsr and bias Asr for the s-th neuron in the r-th layer.	
2. While the model architecture improves, do the following: - If it is not the first execution, attenuate previous parameters. - For t=1,…,N, perform necessary operations on each neuron. - Assume F denotes input and G denotes system output. - Calculate Bsr influenced by the previous layer’s activations:	
     Ajr=σ∑t(wstrBsr−1+Asr)	
- Matrix form:	
     Bl=σ(wrBr−1+Ar)	
- Call the base learner and update the activation function Bl by recalculating layer-wise from F→G^.	
- Loss function:	
     Z=12n∑u∥y(u)−BL(u)∥2	
where n is the number of training samples, v=y(u) represents ground truth samples, M is the total layers, and BL(U) is the output vector. - Find the weight vector w with minimal error via partial derivatives of Wstr.	
- Update weight to reduce errors.	
Output: The model with trained weights and thresholds for each layer.	

The Minimum Redundancy Maximum Relevance (MRMR) algorithm is used as a feature selection approach. Then the classification is performed to classify into eight classes.

Results and discussion

Experimental setup

The experimentation is carried out in Jupyter Notebook with Python 3 language and with 20 GB RAM GPU. The base learners (pre-trained on existing data) are loaded, with lower layers being frozen. Regarding training, since deterministic features are represented by top layers, these are trained for the representation of specific feature sets. The architecture framework is trained with the images extracted from the HyperKvasir dataset. We extracted labeled images from eight representative classes of the HyperKvasir dataset and trained the proposed model with these data. The model is then tested on the Kvasir v1 and Kvasir v2 datasets. The features are extracted in 30 iterations with 20 epochs each. The model is tested in a five-fold cross-validation. The quantitative results are analyzed on the basis of 8,000 observations. The model is trained with 30 epochs and 32 and 64 batch sizes.

Datasets

We have used two GIT datasets for testing purposes. Kvasir v1 and Kvasir v2. Both of these have data equally distributed in eight different classes. These datasets are obtained from Simula datasets. These are publicly accessible and are available online at Simula datasets (Pogorelov et al., 2017; Borgli et al., 2020). These datasets are selected to be used in the study due to its specific focus on endoscopic procedures and vast coverage of gastrointestinal conditions.

Kvasir v1

The Kvasir v1 includes clinical and annotated GIT images that are marked by experienced endoscopists. It includes eight different classes that cover both anatomical landmarks and pathological findings of the GI tract. Anatomical landmarks are features that represent exact locations within the GIT recognizable through endoscopic procedures. It serves as a basis of reference point to diagnose and test for a specific finding. The pathological finding includes the diagnosis classes that are used to classify diseases. Examples include dyed lifted polyps, ulcerative colitis and esophagitis, etc. Figure 3 gives information about anatomical landmarks and pathological findings.

Figure 3 Kvasir v1 dataset with anatomical landmarks and pathological findings (Siddiqui et al., 2024).

Kvasir v2

The Kvasir v2 has 8,000 images uniformly distributed in eight classes. Classes are organized into anatomical landmarks and diagnosis classes are named research findings. Moreover, it also covers the image classes that correspond to endoscopic polyp removal. Images are annotated and sorted by specialized endoscopists that include various classes including endoscopic procedures, anatomical landmarks, and pathological findings. Figure 4 represents Kvasir v2 sample images with anatomical landmarks and pathological findings. The tiny green boxes in some of the images show the location details of the endoscope insertion point inside the duodenum and ileum. This procedure is done by an electromagnetic imaging system (Scope Guide, Olympus Europe) which is performed by an endoscope. The resolution of the image ranges from 720×576 pixels to 1,920 × 1,072 and is sorted into folders named according to the content.

Figure 4 Sample images from the Kvasir v2, showing eight classes with anatomical landmarks and pathological findings.

Green box indicate endoscope insertion points (Siddiqui et al., 2024).

Evaluation metrics

The proposed framework is evaluated on the basis of performance indicators of accuracy, recall, and precision values (Shaga Devan et al., 2022; Grandini, Bagli & Visani, 2020). The predicted outcome of all individual models can be correct or incorrect, in other words, true or false respectively. Therefore, there might be four states of the classification architecture model (Naz et al., 2021): 1. True positive (TPosit).

2. True negative (TNegat).

3. False positive (FPosit).

4. False negative (FNegat).

Below-mentioned equations represent these parameters. Accuracy is given by Eq. (8):

(8) Acc=(TPosit)+(TNegat)TPosit+TNegat+FPosit+FNegat.

Precision is given by Eq. (9):

(9) Prec=(TPosit)TPosit+FPosit.

Equation (10) represents recall:

(10) Rec=(TPosit)TPosit+FNegat.

The average of precision and recall is given by F1-score (Shaga Devan et al., 2022). F1 score is denoted by Eq. (11):

(11) F1=2×Prec×RecPrec+Rec.

The ensemble model is trained by training the two base models using the HyperKvasir labeled images of the eight classes. The batch size, epochs trained, optimizer used, training time, learning rate, and other hyperparameters of the base learners are adjusted to fine-tune the model.

Classification performance evaluation

The performance evaluation indicates that the model outperforms many existing approaches. Table 2 represents the size, depth, parameters and corresponding accuracy values of four different base models.

Table 2 Overview of individual base learners.

Individual base models	Size (MB)	Accuracy (%)	Depth	Parameters (M)	
Inception V3	92	92.1	159	25.6	
ResNet-50	98	93.7	–	23.8	
NasNet-Mobile	23	91.9	389	5.3	
EfficientNet	29	93.3	132	5.3	
GoogleNet	20	93.3	22	5	
DenseNet-121	32	92.21	121	8	

From the table, it is clearly evident that ResNet-50 and EfficientNet have the highest accuracy. EfficientNet has shown to be superior in terms of fewer parameters and lower size (29 MB) compared to many other models, including ResNet-50. ResNet-50 has 23.8 million parameters while EfficientNet gives nearly same results with just 5.3 million parameters. So, we prefer EfficientNet over ResNet-50 because EfficientNet achieves better accuracy with fewer parameters, is more computationally efficient and typically delivers better performance-per-computation. Alongside, we selected NasNet-Mobile to be used with proposed ensemble model as the GIT images may be processed on limited resource devices in medical settings. NASNet-Mobile is considered to be computationally very efficient while maintaining high accuracy. Though ResNet-50 is giving higher accuracy than NasNet-Mobile but due to its large size and more learnables, its training time becomes very high as compared to NasNet-Mobile. So, we preferred NasNet-Mobile over ResNet-50.

Implication of transfer learning networks and statistical analysis

The proposed model incorporates transfer learning by utilizing NasNet-mobile and EfficientNet models to ensemble based on the weighted average ensemble technique. The results are comparatively very satisfactory (93.3% to 98.45% ), hence, a notable difference in percentage accuracy. It is also seen from experimentation that the results of the NasNet-Mobile are more accurate as part of individual fine tuned learner, trained with HyperKvasir dataset than the same used with transfer learning. Figure 5 gives the confusion matrix of the two models as individual base learners.

Figure 5 Confusion matrix of EfficientNet and NasNet-Mobile classification models.

The figure shows that the EfficientNet performed best overall, among the top scorers of all applicable neural network architectures models listed above: With 5.3 M parameters, it is giving us 93.3% accuracy. NasNet-Mobile was also applied, it gives around 92% accuracy and is similar to the ResNet-50 network, except training time. The same model gives recall and precision both with better results when used individually but still, we preferred to use this model with transfer learning. The reason is the impact of training time as the training takes much more time when training the model from scratch thus its applicability is not feasible in real-world scenarios where the dataset is large and fuzzy. We found out after experimentation that each of these models predict better classification than most of the existing STA models. Moreover, to address variances of the deep learning frameworks, the results of different individual model training can be combined on the basis of identified discriminative features.

Evaluation of ensemble learning model

The proposed framework is evaluated on test datasets Kvasir v1 and v2 with different batch sizes and multiple epochs. The model has 10.3 million parameters. The quantitative results are presented in Table 3.

Table 3 Results comparison with different epochs and batch sizes on Kvasir v1 and Kvasir v2.

Test dataset	Training dataset	Batch size	Epochs	Exec. time	Validation split	Accuracy	Precision	Recall	F1 score	AUC score	Cohen Kappa score	
Kvasir v1	HyperKvasir	16	30	5 s/step	0.2	0.984	0.99	0.97	0.97	0.97	0.94	
Kvasir v2	HyperKvasir	32	30	10 s/step	0.2	0.978	0.98	0.97	0.96	0.97	0.92	

It is evident from the table that with batch size 32 and 30 epochs, accuracy reaches at 98.45% for Kvasir v1. For Kvasir v2, we split batch size to 32 as it is more fuzzy and comparatively larger dataset. The model achieves 97.85% for Kvasir v2. The proposed work represents two different classifiers with comparatively accurate outcomes, selected as the individual learners among the four CNN models for analysis namely the NasNet-Mobile and Efficient-Net architectures. The accuracy is considered to be the primary parameter for selection among all network architectures and weights are set to be 0.4 and 0.6 for EfficientNet and NasNet-Mobile, respectively. Lastly these weights are implemented in the final ensemble architecture. So, after these steps, we achieved an accuracy of 98.45% for Kvasir v1 and 97.83% for Kvasir v2. Table 4 indicates the quantitative comparison of the proposed model with other hyperparameters optimized fine-tuned STA models.

Table 4 Percentage accuracy on Kvasir v1 and v2 datasets.

Model	Kvasir v1 accuracy (%)	Kvasir v2 accuracy (%)	
NasNet-Mobile	94.53	93.21	
EfficientNet	92.28	90.28	
Proposed ensemble model	98.45	97.83	

Figure 6 indicates the confusion matrix of the ensemble model. The confusion matrix represents increased percentage accuracy of 97.83%. The proposed model outperforms in accuracy as compared to the other pretrained models. It also provides leverage of using knowledge gained from transfer learning instead of training the model from scratch. It reduces training time and computation cost as well.

Figure 6 Confusion matrix of the proposed model.

The statistical results are evaluated and analyzed. Figure 7 indicates quantitative trends that with changing epochs, accuracy increases and validation loss decreases. This line chart indicates the changing trend of accuracy, validation and training loss with increasing number of epochs. The graph indicates that after 15 epochs, execution time increases at a very rapid rate. At 30 epochs, training loss reaches at a minimum value and validation loss increases. Validation loss has irregular pattern but decreases drastically after five epochs.

Figure 7 Chart representing accuracy, validation loss, training loss, and time with changing epochs.

Comparative experiments

A comparative survey is performed by comparing different classification methods with the proposed architecture. Table 5 gives the quantitative results of the experimentation. From the above table, it is evident that the proposed model outperforms existing techniques in terms of accuracy and sensitivity.

Table 5 Percentage accuracy comparison between STA approaches and the proposed method.

Previous Studies	Accuracy	Sensitivity	Specificity	AUC	
Polyp detection and segmentation (Pozdeev, Obukhova & Motyko, 2019)	88	93	82	–	
Pretrained individual classifiers (Bour et al., 2019)	87.10	87.10	93.00	–	
Polyp identification based on Convolutional Neural Networks (Zhang et al., 2016)	85.90	87.60	–	86.00	
Deep learning for colorectal polyps (Ribeiro, Uhl & Häfner, 2016)	90.19	95.16	74.19	–	
Colonoscopy based RGB technique for cancer diagnosis (Min et al., 2019)	78.40	83.30	70.10	–	
Classification of GIT disorders based on deep learning (Fonollá et al., 2019)	90.20	90.10	90.30	97.00	
Histology based computer-aided diagnosis (Tan et al., 2016)	90.19	95.16	74.19	–	
Pretrained deep classifier (Balagourouchetty et al., 2019)	96.70	96.60	99.90	99.70	
HOG based feature selection (Godkhindi & Gowda, 2017)	88.56	88.77	87.35	–	
Skin cancer identification based on lesion identification and segmentation (Zhu, Zhang & Xue, 2015)	85.00	83.00	82.54	–	
Proposed model	98.45	99.0	98.3	99.7	

Conclusion and future work

In this study, we developed and evaluated a transfer learning-based framework for GIT image classification. The framework leveraged pre-trained deep learning models NasNet-Mobile and EfficientNet, which were fine-tuned on HyperKvasir datasets to improve classification accuracy while minimizing the need for extensive labeled data. The model was tested on Kvasir v1 and v2 datasets. This is done to produce a more robust model. We employed two pre-trained models, NasNet-Mobile and EfficientNet, to create an ensemble model. NasNet-Mobile achieved the best accuracy, with 94.53%. Ultimately, a single architectural model proved insufficient for effective prediction, while the ensemble model attained accuracy rates of 98.45% on Kvasir v2 and 97.83% on Kvasir v1, respectively. The proposed method, based on weighted averaging, outperformed base learners acting individually. However, ensemble network architectures present some drawbacks, including approximately 34% longer training times and a more complex architecture, comprising 66 million hyperparameters. The accuracy of the model varies significantly based on the characteristics of the Kvasir v1 and v2 datasets and can be notably affected by the segmentation methods applied before classification.

Performance was evaluated using a weighted averaging framework, where individual models with higher accuracy received greater weights; NasNet, with a weight of 0.6, outperformed EfficientNet, which was assigned a weight of 0.4. This approach yielded accuracies of 98.45% and 97.83%, surpassing those of the individual pre-trained base models, thus demonstrating that weighted ensemble models provide more accurate results than their counterparts. The limitations of the work include its applicability to some fuzzy and unbalanced datasets. For unbalanced datasets, we might need to include segmentation for better results. Moreover, the proposed model might not perform well on unbalanced data. The future work includes applying explainable AI (XAI) in interpreting the results of the proposed approach. It includes more transparency in gained results.

We would like to express our gratitude to Dr. Omer Chughtai for his valuable contributions and guidance. We also extend our sincere thanks to Dr. Muddassar Raza for his continuous effort and support.

Additional Information and Declarations

Competing Interests

The authors declare that they have no competing interests.

Author Contributions

Samra Siddiqui conceived and designed the experiments, performed the experiments, analyzed the data, performed the computation work, prepared figures and/or tables, and approved the final draft.

Junaid Ali Khan analyzed the data, prepared figures and/or tables, authored or reviewed drafts of the article, and approved the final draft.

Shabbab Algamdi performed the experiments, authored or reviewed drafts of the article, and approved the final draft.

Data Availability

The following information was supplied regarding data availability:

The code is available at GitHub and Zenodo:

- https://github.com/SamraFawad/Ensemble-Model-code.git.

- Siddiqui. (2025). Ensemble Model Code. Zenodo. https://doi.org/10.5281/zenodo.15055211.

The Kvasir data is available at https://datasets.simula.no/kvasir.

The HyperKvasir Dataset is available at OSF: Hanna Borgli, Vajira Thambawita, Pia H. Smedsrud, Steven Hicks, Debesh Jha, Sigrun L. Eskeland, Kristin Ranheim Randel, Konstantin Pogorelov, Mathias Lux, Duc Tien Dang Nguyen, Dag Johansen, Carsten Griwodz, Håkon K. Stensland, Enrique Garcia-Ceja, Peter T. Schmidt, Hugo L. Hammer, Michael A. Riegler, Pål Halvorsen, and Thomas de Lange: “HyperKvasir, a comprehensive multi-class image and video dataset for gastrointestinal endoscopy”, Springer Nature Scientific Data, Vol. 7, 2020 DOI: https://doi.org/10.1038/s41597-020-00622-y.

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
