# Peer review of "Deep ensemble learning for gastrointestinal diagnosis using endoscopic image classification"

_PeerJ Computer Science, doi:10.7717/peerj-cs.2809_

## Round 0.1 · original submission · Major Revisions

Based on the referee reports, I recommend a major manuscript revision. The author should improve the manuscript, carefully consider the reviewers' comments in the reports, and resubmit the manuscript.

Reviewer 1 ·

Basic reporting

- There are grammatical errors in the language of the article, it should be reviewed again.
- A deep ensemble model based on transfer learning is proposed in the article. In the abstract section, information should be provided about the architecture of the model used and the training process, even if it is just a few sentences.
- Is GIT an abbreviation for Gastrointestinal (line 15) or gastrointestinal tract (line 32)?
- In the introduction section, information should be provided about the details of the proposed model before the contributions, what are the main steps of the method? It should be presented in this section.
- The 2nd contribution is not a contribution of the proposed method. It is one of the steps of the method. The contributions and their motivations should be explained more clearly.
- The number of previous studies reviewed for GIT should be increased. The related studies section is insufficient.
- It would be appropriate to show Figure 1 horizontally or by showing the steps from top to bottom.
- Only Acc was used when evaluating the performances of the algorithms. Therefore, there is no need to give the unused metrics in Section 4.3, they should be removed.

Experimental design

- When comparing ResNet and NASNet, training times were mentioned, what are the training times? This information is not included in Table 2.
-Epoch numbers were chosen differently for Kvasir v1 and Kvasir v2 datasets. Is the comparison made fair in this case?

Validity of the findings

- Why were NASNet and EfficientNet used?
-What are the wt1 and wt2 values?
- The value of 99.2% was not seen in Table 2.
- What do the columns in Table 2 represent, what were the Size, Depth, and parameters (M) values ​​used for?
- The analyses related to the tables given throughout the article are incomplete, there are almost no explanations. Only tables are given in the subsections, analyses and explanations should definitely be added or increased.

Additional comments

I have reviewed the article titled "Deep Ensemble Learning for Gastrointestinal Diagnosis Using Endoscopic Image Classification" in detail. There are significant deficiencies in the paper, it is unacceptable in its current state. It should be reviewed well, considering the comments I wrote under three headings.

Reviewer 2 ·

Basic reporting

All comments have been added in detail to the last section.

Experimental design

All comments have been added in detail to the last section.

Validity of the findings

All comments have been added in detail to the last section.

Additional comments

Review Report for PeerJ Computer Science
(Deep ensemble learning for Gastrointestinal diagnosis using endoscopic image classification)

1. Within the scope of this study, classification tasks were performed on gastrointestinal images using a proposed deep learning-based model.

2. In the introduction, Gastrointestinal disorders and the importance of the subject, as well as the contributions of the study, are mentioned at a basic level but at a certain level.

3. The literature review section is very limited and should definitely be detailed. Literature can be detailed, especially in relation to the studies conducted with the Kvasir dataset used in the study. In addition, table-1 can be detailed by adding certain columns such as the number of studies in the literature and the pros and cons of the studies, their originality, and preprocessing steps.

4. When the proposed architecture using the NasNet and EfficientNet models is examined in detail, it is understood that it has a certain level of originality and has the potential to contribute to the literature. In addition, the preprocessing steps performed on the dataset are also sufficient.

5. The proposed model is at a certain level. However, there are still parts that need to be clarified. In this context, although there are many different and more up-to-date deep learning models that can be used in the literature, it should be explained more clearly why EfficientNet and NasNet models were preferred for the ensemble model and/or whether different experiments were performed.

6. In the study, two different versions of Kvasir were preferred as datasets. Although it is important to use more than one dataset, it should be detailed why Kvasir versions were preferred and not different open source datasets related to Gastrointestinal.

7. In the Experimental setup section, the toolbox and framework information used in the application phase should be detailed. In addition, the hyperparameter selections and details used should be specified.

8. Although certain metrics were obtained in the evaluation metrics section, there are missing metrics for the correct analysis of the results. For this reason, all missing metrics, especially precision, recall, f1-score, ROC (Receiver operating characteristic) curve, AUC (the area under the ROC curve) score, Cohen’s Kappa score and MCC (Matthews Correlation Coefficient) score, must definitely be obtained.

As a result, the study is at a certain level in terms of the subject and the proposed model. However, all the sections mentioned above should be addressed completely.

·

Basic reporting

Manuscript ID 109415v1
This paper is related to reviewing the manuscript titled "Deep ensemble learning for Gastrointestinal diagnosis using endoscopic image classification"
This research proposes a deep ensemble model leveraging transfer learning for accurate classification of GIT disorders in endoscopic images, focusing on identifying and classifying diseased regions.
Firstly, the presented study is successful in terms of good literature research and evaluation results.

Experimental design

1) Use abbreviations after the first use in the text, in the abstract and throughout the paper.
2) The conclusion section really needs to be improved
3) The resolution of the figures giving should be increased.
4) Clean the paper of English spelling and punctuation errors

Validity of the findings

1) Use abbreviations after the first use in the text, in the abstract and throughout the paper.
2) The conclusion section really needs to be improved
3) The resolution of the figures giving should be increased.
4) Clean the paper of English spelling and punctuation errors

Additional comments

My decision is minor revision. I would like to inform you that if all the requested items are not completed in this revision, my decision will be to reject the application in the second round. Otherwise, I do not see any harm in publishing the manuscript once the above revisions are made.

---

## Round 0.2 · Major Revisions

Kindly revise the manuscript as per reviewer1 suggestions and resubmit it.

Reviewer 1 ·

Basic reporting

-There are many grammatical errors in the paper. The language of the article should definitely be revised.
- A paragraph providing details of the proposed method should be given above the contributions in the introduction section. The difference between this method and the studies reviewed between lines 53-65 and the motivation and purpose of the method should also be added.
- The literature review section should be updated by adding the most recent (2025) studies on gastrointestinal diagnosis to the related studies section. Example doi: 10.1016/j.bspc.2024.106847, 10.1109/ACCESS.2024.3522009.
- Acc values ​​should be given in the same format in Table 1: percentage or decimal format should be selected.

Experimental design

- Why were NasNet-Mobile and EfficientNet preferred?
- Increasing the number of compared algorithms in Table 2 will enrich the scope of the study.
- What are the parameter values ​​of the algorithms?

Validity of the findings

- The method is explained under subheadings in section 3. This approach is correct. However, in the introduction of section 3, the steps in Figure 1 should be explained with one sentence each, allowing a transition to the next subsections.
- What should be the future studies to solve the limitations of the study, should be given in the last section.

Reviewer 2 ·

Basic reporting

All comments have been added in detail to the last section.

Experimental design

All comments have been added in detail to the last section.

Validity of the findings

All comments have been added in detail to the last section.

Additional comments

Review Report for PeerJ Computer Science
(Deep ensemble learning for Gastrointestinal diagnosis using endoscopic image classification)

Responses to reviewer comments and changes made to the paper are generally at an appropriate level.

---

## Round 0.3 · accepted · Accept

The author has addressed the reviewer
comments properly, so I think the manuscript can be published.

Reviewer 1 ·

Basic reporting

-

Experimental design

-

Validity of the findings

-

Additional comments

The authors have made all necessary corrections and the article can be accepted as it is.